# Study Protocol: Strategies and Techniques for the Rehabilitation of Cognitive and Motor Deficits in Patients with Multiple Sclerosis

Ornella Argento [1,*], Chiara Piacentini [1], Michela Bossa [1], Carlo Caltagirone [2], Andrea Santamato [3], Vincenzo Saraceni [4] and Ugo Nocentini [1,5]

1   Behavioral Neuropsychology Laboratory, I.R.C.C.S. "Santa Lucia" Foundation, 00179 Rome, Italy; c.piacentini@hsantalucia.it (C.P.); m.bossa@hsantalucia.it (M.B.); u.nocentini@hsantalucia.it (U.N.)
2   Scientific Direction, I.R.C.C.S. "Santa Lucia" Foundation, 00179 Rome, Italy; c.caltagirone@hsantalucia.it
3   Rehabilitation Centre, Physical Medicine and Rehabilitation Section, OORR Hospital, University of Foggia, 71100 Foggia, Italy; andrea.santamato@unifg.it
4   Scientific Direction, "Filippo Turati" Foundation, Rehabilitation Centre, 51100 Pistoia, Italy; vincenzosaraceni@gmail.com
5   Department of Clinical Sciences and Translational Medicine, University of Rome "Tor Vergata", 00133 Rome, Italy
*   Correspondence: o.argento@hsantalucia.it; Tel.: +39-06 51501191

**Abstract:** MS clinical features vary between patients. In approximately 60% of cases, cognitive deficits are associated with motor disabilities, with consequences on both walking and maintaining balance and cognitive efficiency. Multimodal programs are very infrequent for MS patients and cognitive rehabilitation is not provided by the Italian health system, which only favors access to motor rehabilitation. Dual-task studies showed how motor and cognitive skills are closely associated. Therefore, physiotherapy exercises may favor an indirect improvement in cognition. The aim of this study is to understand which rehabilitative approach may increase both cognitive and motor efficiency, avoiding the waste of time and resources. In this multi-site single-blind parallel controlled clinical trial, we will compare three rehabilitative approaches: cognitive training, motor training and combined cognitive–motor training. We also aim to evaluate: whether self-perception and objective improvement correspond; the impact of each rehabilitation program on patients' QoL, mood and self-perception; and long-term effects. A total of 60 patients will be randomly assigned to one of the three treatments for two 45-min sessions/week for 12 weeks. All participants will undergo a complete cognitive, motor, clinical assessment together with mood, self-perception, and QoL questionnaires before, immediately after and 6 months after the training period.

**Keywords:** study protocol; rehabilitation; memory; multiple sclerosis

## 1. Introduction

Multiple sclerosis (MS) is a chronic inflammatory, demyelinating and degenerative disease of the central nervous system (CNS): the second-most frequent cause of permanent disability in young adults. The clinical characteristics of MS are extremely variable from one patient to another. In approximately 60% of cases, motor disabilities are associated with cognitive deficits. Concerning motor efficiency, the main sources of disability are walking and maintaining balance [1]. From a cognitive point of view, the main and most frequent emerging deficits concern memory, information processing speed and attention [2]. However, all cognitive functions may be involved even if in a lower percentage of cases. Both motor and cognitive disability emerge even in the early stages of the disease and have a serious impact on patients' and caregivers' quality of life. Between them, the most disabling disorders perceived by patients with MS (pMS) are walking deficits, as they most

directly affect the ability to efficiently perform even simple daily tasks. For this reason, 65% of pMS consider walking ability rehabilitative intervention a priority [3].

On the other hand, cognitive impairments are often underestimated by pMS, even if they interfere with daily life activities [4,5]. Memory impairment, in example, has a particularly negative impact on social and relational patients' life [6] and on depression levels. To date, both disease-modifying treatments and specific drug therapies have shown little or no impact on the cognitive deficits complained of by pMS [7,8]. Similarly, motor deficits, once established, are not easily modifiable by pharmacological treatments. We know that rehabilitation strategies improve various functional aspects in pMS [9,10]: an effect probably connected to the structural and functional modifications that occur in the CNS of the treated patients.

The research on rehabilitation programs for MS-related cognitive impairments has become increasingly important: these programs could make patients more autonomous in the management of work and home duties and lighten caregivers' burden [11,12]. Particular efforts have been directed towards the development of rehabilitation programs for the empowerment of memory: MEMREHAB [13,14], RehaCom [15–17] and the modified-Story Memory Technique (mSMT) [18] have shown encouraging results.

The most recent reviews of the literature tried to identify the most suitable rehabilitation tools for each cognitive function in pMS. However, they encountered several difficulties in comparing the results across studies because: some are focused on single-domain rehabilitation, for example the impaired memory function [14], whereas others have set the rehabilitation pathways more broadly, also rehabilitating other connected cognitive functions [17]. This is the case for intensive cognitive training for attention/speeded information processing, executive functions, and memory compared to an aspecific psychological intervention [17]. Furthermore, more recently, interesting results have emerged on the interaction between deficits in cognitive functions and motor impairments [19–21]. In particular, it seems that the real disabling effect of MS on patients may be better understood when we consider not only cognitive and motor deficits separately, but also the so called "dual-task effect" describing the mutual interference generated by both deficits each other [21]. This has opened up the possibility that treating cognitive and motor impairments at the same time could be even helpful in enhance the mutual influence of cognition and motor circuits each other.

In this regard, some researchers suggested that a better rehabilitation outcome can be obtained by combining cognitive with motor rehabilitation exercises, than when cognitive and motor functions are treated separately [22]. This makes it difficult to establish which is the most valid rehabilitation approach for the improvement of cognitive deficits (and in particular memory ones) in patients with MS.

In order to increase the possibility of treating cognitive deficits in MS patients, it could be useful to have comparative data on the efficacy of cognitive rehabilitation performed alone with respect to motor rehabilitation alone and with a combined cognitive and motor rehabilitation program.

## 2. Materials and Methods

### 2.1. Aim, Design and Setting of This Study

This study has the following primary objectives:

1. Compare three forms of rehabilitation—cognitive, motor and combined (cognitive–motor)—in three groups of MS patients.
2. Check whether a combined rehabilitation approach can induce a significantly greater improvement on memory efficiency of MS patients, than cognitive rehabilitation performed alone.

Secondary objectives:

1. Evaluate whether self-perception and objective improvement correspond.
2. Evaluate the impact of each rehabilitation program on patients' overall disability.
3. Monitor the effects of the three rehabilitation conditions after 6 months.

This study will be a single-blind parallel controlled clinical trial, in which each subject will be randomly assigned to a single treatment. The study protocol will be divided into an initial evaluation by standardized tools of clinical, cognitive, emotional, quality of life and functional self-perception state, at time 0 (T0). Subsequently, patients will be assigned to one of the three conditions and will undergo rehabilitation treatment for a total of 12 weeks (two sessions each week). At the end of the treatment, each patient will undergo an overall re-evaluation (T1) with parallel forms of the tools used at T0. After 6 months, all patients will be submitted to a further overall re-evaluation as a follow-up measurement of each treatment (T2). The examiner carrying out pre- and post-treatment evaluations will be different from the one carrying out the rehabilitation process in order to avoid any individual bias. Figure 1 shows the study flowchart.

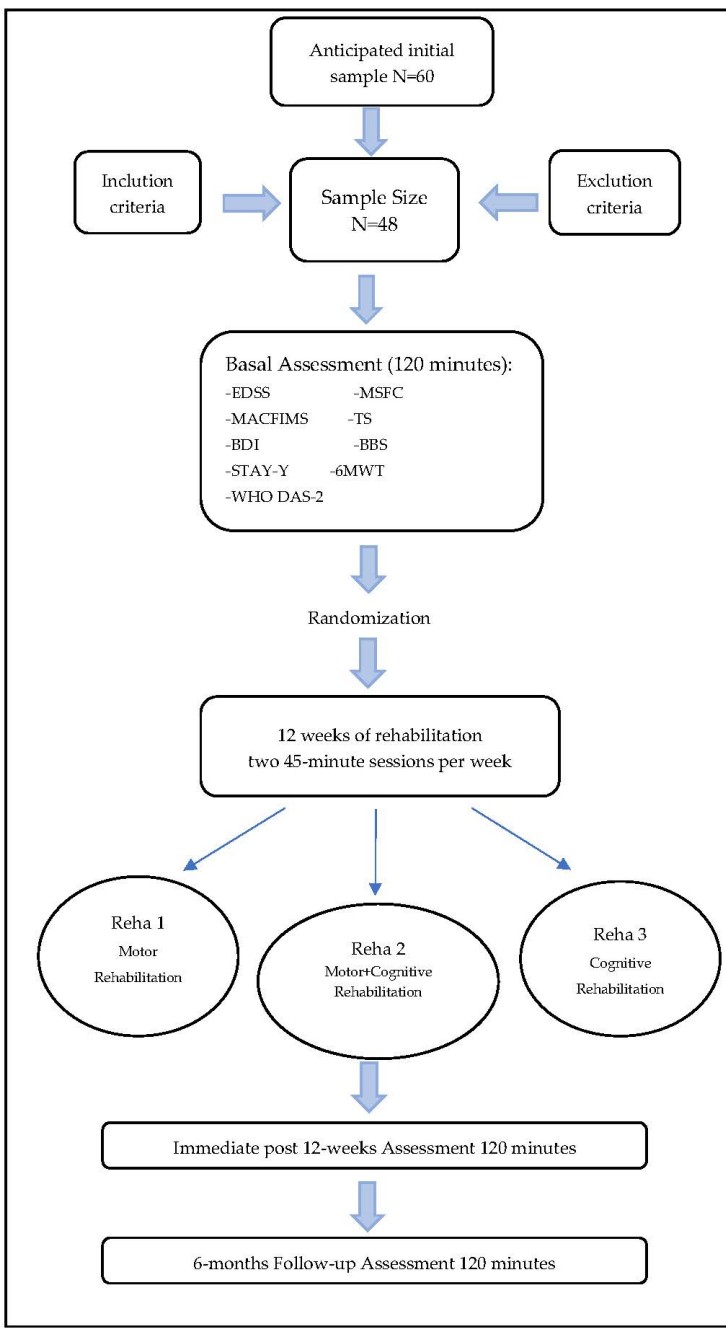

**Figure 1.** Study flowchart.

Patients in charge of both the "Santa Lucia" Neuroscience and Rehabilitation Foundation (Rome) and the "Filippo Turati" Foundation (Rome) will be recruited. All evaluations and rehabilitation training will be conducted at the laboratory and in the rehabilitation sites of the two centers by adequately trained neurologists/physiatrists, psychologists and physiotherapists.

For the 12 weeks following randomization, each patient will undergo one of three rehabilitation programs:

- Reha 1: is composed by a computerized cognitive rehabilitation training using four sessions of the verbal memory set of the RehaCom program [15–23]. Each patient will carry out two 45-min sessions/week for 12 weeks;
- Reha 2: is composed by a combined rehabilitation program with the use of the four verbal memory sessions of RehaCom together with a motor rehabilitation program. Each patient will carry out one 45-min session of cognitive rehabilitation and one 45-min session of motor rehabilitation each week for 12 weeks;
- Reha 3: is composed of two 45-min sessions of motor rehabilitation training each week for a total duration of 12 weeks.

### 2.2. Recruitment and Screening of Participants

Participants in this study will be selected by the treating neurologists. Subsequently, they will be contacted via telephone by the researchers. All patients will be informed about the purposes of the research and will have to sign an informed consent before undergoing any type of evaluation. Patients who will meet the inclusion criteria will be assigned to one of the three rehabilitation conditions using a stratified randomized sampling strategy in order to control variables such as disease course or sex, which could make the groups not comparable. The inclusion and exclusion criteria are summarized in Table 1.

**Table 1.** Inclusion and exclusion criteria for the enrollment of patients.

| Inclusion Criteria | |
|---|---|
| **Diagnosis** | Diagnosis of MS defined according to McDonald's diagnostic criteria revisited in 2011 |
| **Age** | 18–65 years |
| **Phenotype** | RRMS or SPMS |
| **Language** | Italian mother tongue |
| **EDSS** | <6.0 |
| **Exlcusion Criteria** | |
| **Pathologies** | Neurological or psychiatric conditions other than MS that can interfere with cognitive functioning |
| **Neurological history** | Clinical relapses in the three months prior to enrollment, neurological pathologies other than MS and severe enough to interfere with cognitive functioning, and clinical relapse or disease activity highlighted on MRI magnetic during the treatment period |
| **Severe mental illness** | Psychiatric disorders severe enough to interfere with cognitive functioning |
| **Medications** | Steroid therapy in the 3 months prior to enrollment |
| **Motor limitations** | Dysfunction of the upper limbs (paralysis or tremor) that do not allow to hold the PC mouse |
| **Sensory limitations** | Visual acuity impaired enough not to allow reading of the instructions to the various tests |

### 2.3. Assessment Procedures

At T0 all patients will undergo the following assessments:

- Accurate collection of personal, anamnestic and clinical information;
- Expanded Disability Status Scale (EDSS) [24]: is a rating scale administered by a trained physician, aimed at evaluating functional CNS sub-systems. EDSS is used

also to identify disease progression in MS patients and to evaluate the effectiveness of therapeutic interventions in clinical trials. It consists of an ordinal classification system ranging from 0 (normal neurological status) to 10 (death for MS) in increments of 0.5 (when EDSS 1 is reached). The lower values of the EDSS scale measure deficits based on neurological examination, while the upper range of the scale (>EDSS 6) measures the handicaps of MS patients [25]. The determination of the 4–6 interval of the EDSS strongly depends on aspects concerning walking [25].

- Neuropsychological evaluation through the Minimal Assessment of Cognitive Functioning in Multiple Sclerosis battery (MACFIMS) [26], in the Italian version [27]. This battery consists of seven tests:

  ○ The California Verbal Learning Test-II (measurement of verbal learning and memory; CVLT-II) [28,29] requires the learning of a 16 words list (made up of 4 words from four different categories) read by the examiner across five trials. After each trial the subject has to recall as many words as possible in any order; the number of words correctly recalled for each trial is recorded. The total Immediate Recall (IR) score is the sum of the words correctly recalled across the five trials, ranging from 0 to 80. After having performed some interference tests the patient has to repeat the previously learned words (Delayed Recall; DR).

  ○ The Brief Visuospatial Memory Test—Revised (visuospatial memory test; BVMT-R) [30] consists of three consecutive trials in which the subject views for 10 s a display of six non-iconic figures, arranged in a 2 × 3 matrix. After that the subject has to reproduce each figure in the correct locations on the page. Patient is asked to reproduce the figures again after a 25-min interval without any further exposure to the stimuli. Scoring is based on the subject's reproduction accuracy and location of each figure [31]. Each figure reproduced can receive 0 to 2 points score [32]. The Immediate Recall (IR) corresponded to the scores' sum obtained across the three trials while the Delayed Recall (DR) is the score obtained in the 25-min delayed trial. The BVMT-R offers six equivalent alternative forms [33].

  ○ The Symbol Digit Modalities Test (information processing speed; SDMT; oral version) features a series of nine symbols, each of which is paired with a single-digit number in a pattern at the top of a sheet. The rest of the page has a pseudo-randomized sequence of these symbols. The patient is asked to say the digit associated with each symbol as quickly as possible in 90 s. The SDMT was originally designed for both written or oral responses, but the expert group recommended oral administration with MS patients to minimize complaints due to upper limb weakness or incoordination. The dependent variable is the total number of correct answers in 90 s [33,34].

  ○ The Benton Judgment of Line Orientation test (measures the accuracy of spatial orientation judgments; BJLO) [35] requires subjects to identify the angle defined by two stimulus lines among those reported in a visual series of lines spanning 180 degrees. The dependent variable of the BJLO is the total number of correct answers [33].

  ○ The Controlled Oral Word Association Test (measure of phonemic fluency; COWAT) gives subjects three 1-min time intervals each to generate as many words as possible beginning with three different input letters. Given that the subject's performance is strongly influenced by the efficiency and speed of the search into one's own lexicon, the COWAT cannot be considered a test of "pure" language. The total score is the total number of words generated in all three tests. Two equivalent alternative forms are available [33].

  ○ The Delis–Kaplan Executive Function System Sorting Test (D-KEFS ST) is a composite measure of: concept formation skills, specific problem-solving skills for verbal/non-verbal aspects and ability to explain the abstract ordering of

concepts [36]. It is able to explore reasoning, categorization skills, problem solving, abstraction, flexibility of thought and conceptual training skills, and it also provides good validity [37] and adequate reliability [38]. The test involves the presentation of six mixed cardstocks having different perceptual and semantic characteristics. The participant is asked to divide the cards into two groups (categorization), with three cards each, based on objective criteria and to describe the concepts used to generate each categorization (description). Each of the two sets of cards has a maximum of eight types of categories: three based on the semantic meaning of the printed words and five based on the cards' visuo-spatial characteristics or patterns. The participant has a maximum of 4 min (for each set of cards) to find as many categorizations as possible [39]. The Sorting Score (SS) represents the total number of correct categorizations made by the subject; the Point Score (PS) represents the quality of the description (classifications) made for all the SS.

○ The Paced Auditory Serial Addition Test in 3 and 2 s versions (measure of working memory; PASAT). It takes 20 min to administer and has adequate sensitivity and specificity (approximately 75% and 90%, respectively) in discriminating compromised patients from intact ones [40,41] In this test the patients asks some digit numbers each 3 or 2 s. He/she is asked to sum the number just said with the previous and say each time the correct sum.

- The Beck Depression Inventory (BDI) [42] is a self-report tool that allows you to assess the severity of depression in patients of at least 13 years of age. The test consists of 13 items with a score ranging from 0 to 3 from which a total score is derived. The test was developed as an indicator of the presence and intensity of depressive symptoms at the time of administration. It is useful in assessing depressive mood changes, estimating suicide risk and correctly assessing depressive symptoms in primary prevention, intervention and follow-up.

- The State–Trait Anxiety Inventory Y form (STAI Y) [43] is made up of two subtests of 20 items each with a 4-level response scale of intensity (e.g., from "almost never" to "almost always"). The first subtest evaluates how subjects feel at the time of testing, the second focuses on how subjects generally feel. The two scales refer, respectively, to state anxiety (considered a temporary interruption of the emotional continuum), conceived as a particular experience, a feeling of insecurity, of helplessness in the face of a perceived damage that can lead either to worry or to escape and avoidance; and trait anxiety (considered a relatively stable personality characteristic or a behavioral attitude) which reflects the tendency to perceive stimuli and environmental situations as dangerous or threatening [44].

- The World Health Organization—Disability Assessment Schedule (WHO-DAS 2—Disability level assessment questionnaire) is a tool developed by the WHO in 1998 in order to evaluate the limitations of activities and restrictions on participation experienced by a patient due to its medical condition. The WHODAS-2 is made up of 36 items that evaluate the functioning and the disability of the subject in a time window of 30 days [45]. The questionnaire covers 6 domains: cognitive functions (6 items), mobility (5 items), self-care (4 items), interaction with others (5 items), daily life activities [domestic activities (4 items), work/school (4 items)], participation in social life (8 items). The answer options range from 0 (no difficulty) to 5 (total difficulty) [46].

- The Multiple Sclerosis Neuropsychological Questionnaire (MSNQ) [47] is a brief self-administered test with 15 questions that reflect neuropsychological competence during activities of daily living. Each item is rated on a 5-point Likert scale, which goes from 0 (never, never happens) to four (very often, very seriously). A total score is obtained from the sum of the points of each single item.

- The Multiple Sclerosis Functional Composite (MSFC) is a functional assessment consisting of three tests:

1.  The Timed 25 Foot Walk (T25FW) is a quantitative measure of the function of the lower limbs. The patient is directed to the end of a clearly indicated path and is asked to walk 7.62 m as quickly as possible, but safely. The activity is immediately repeated making the patient return to the initial starting point. If necessary, patients can use an assistive device during the test. Three scores are obtained: two relating to single walks and one given by the averaged time of the two paths [48].

2.  The Nine Hole Peg Test (NHPT) is a simple and relatively quick evidence-based test that measures the functions of the upper limbs (hand and arm) [49]. It consists of a standardized test apparatus consisting of a platform with nine holes and nine pegs to insert. Participants are seated and then asked to insert and then remove the nine pegs from the nine holes, one at a time, as quickly as possible. Both dominant and non-dominant hands are tested twice (two consecutive dominant hand trials, immediately followed by two consecutive non-dominant hand trials) [48]. At the end of the test, 6 scores are obtained: 2 relating to the time taken to perform the task with the dominant hand, 2 scores relating to the time spent with the non-dominant hand and finally two scores representing the average time taken to perform the test with the dominant and the non-dominant hands.

3.  The PASAT (see on the previous page).

-   The Six Minutes Walking Test (6MWT) is a test developed by Balke in 1963 [50], as an index of motor resistance [51]. The test requires the subject to walk for 6 min without interruption, following a path marked by the therapist through the aid of cones that delineate the boundaries. The subject is allowed to rest when necessary [50]. The distance traveled is measured in meters by a special tool: the measuring wheel.

-   The Tinetti Scale (TS) is a clinical tool that allows the assessment of patient's balance and walking performance by assigning an objective score to the motor performance. The Tinetti scale is composed of a section that evaluates static balance, characterized by 9 items with a global score between 0 and 16, and a section that evaluates gait, composed of 7 items with a global score between 0 and 12. Both sections can be scored by the examiner on a 3-point ordinal scale (0 = inability to execute the request; 1 = ability to execute with adaptations or aids; 2 = ability to execute it without adaptation) or 2-point ordinal scale (0 = inability to execute the request; 1 = ability to execute it without adaptation). The patient can obtain a total score between 0 and 28. Thanks to the obtained score, the examiner is able to quantify the risk of falling and has a basis on which to develop personalized rehabilitation programs [52].

-   The Berg Balance Scale (BBS) is one of the most common clinical measures of static and dynamic balance [53]. The BBS includes 14 items that assess a person's ability to maintain balance while performing activities of different difficulty (e.g., going from sitting to standing, picking up an object from the floor, turning 360°). A score is assigned to each item on a 4-point ordinal scale from 0 (=unable to perform) to 4 (= normal performance), accounting for the patient's ability in performing the exercises without any aid. The BBS has a strong inter-rater and test–retest reliability [54].

*2.4. Randomization*

After having completed the assessment procedures, the participants will be randomized to the three rehabilitation arms. The randomization assignment will foresee a stratification by age groups (<30; 30–40; 41–50; 51–65; >65) and will be weighted with respect to the variable phenotype of pathology (each group will be guaranteed a distribution of patients in RR or SP course representative of the incidence of the two courses in the Italian MS population) [55]. This will be a single-blind study.

## 3. Rehabilitation

*3.1. Cognitive RehabilitationRehaCom Modules*
RehaCom Modules

CR training will be performed by the computerized program RehaCom [15–23]. This program will be installed on a laptop with an external mouse attached. The sections of the program used to rehabilitate memory deficits will be:

- *Strategic memory training (LEST)*: The patient will be presented with a pair of words on the PC screen to memorize (the number of words could increase or decrease according to the performance of the subject). The task will be to find the objects that correspond to the memorized words. When the patient recognizes an object, he will have to click on it and repeat the operation until he/she has clicked on all recognized objects. The task will end when all the objects are identified. The correct choice will be marked in green, the wrong one in red. For each group of images presented, the subject will have three possibilities of error, after which the exercise will return to the opening words and the patient is asked to memorize them again. The words to be memorized will change only when all the images are recognized without making mistakes. Patients can use any personal memorization strategies: some will memorize the words according to the presentation order, others by dividing them into semantic categories and others by building stories. Between memorizing words and recognition, participants will be asked to perform a distracting task: move a basket from one side of the screen to the other to collect all the fruits that fall from a tree.
- *Working memory (WOME)*: The patient will be presented with three decks of cards (a deck of poker cards, a deck of colored cards and one characterized by unusual symbols) from which he/she can choose his/her favorite. Then, the program asks the patient to follow some specific indications (e.g., to memorize all the cards presented, to memorize the cards in the presentation order, and to memorize only cards with certain characteristics). In the event that the patient does not perform the exercise correctly, this will be repeated a second time with the same cards. To increase the difficulty of the exercise, a distractor (e.g., a question of general knowledge) may be introduced between the memorization of the cards and their recognition.
- *Figurative memory (BILD)*: Words to be memorized without time limits will appear on the PC screen. Once all the words are memorized, the patient will see a series of figures moving across the screen from right to left. When the patient sees one of the figures, corresponding to the objects memorized before, passing through a red highlighted area, he must press the "ok" button. The number of figures will progressively increase with the improvement of the patient's performance.
- *Verbal memory (VERB)*: The patient will see a short text in which he/she has to memorize names, numbers, objects and events. Subsequently, the subject is asked to answer some questions about what has been read, choosing from four answer options. The difficulty of the task may decrease or increase by varying the length of the passage and the amount of information depending on the patient's performance. Furthermore, the settings of the task can be changed: for example, the exercise can be set in such a way as to have the patient read two passages, one after the other, and then answer questions asked about the first one, or the patient can be asked to answer with open answers.

*3.2. Motor Rehabilitation*

Diversified and specific rehabilitation sessions will be planned for each patient, paying particular attention to motor disabilities of the lower limbs, most commonly observed and complained of by almost all patients with MS. In this regard, exercises will mainly be applied to improve: decreased range of motion, spasticity, balance disorders, coordination problems, reduced postural control, strength deficit, impaired sensitivity, motor difficulties in walking and in the transition from sitting to standing and vice versa.

The proposed exercises are presented in the Supplementary Materials section.

## 4. Statistics and Outcome

### 4.1. Data Analysis

The sample size has been determined based on an a priori power analysis, using the software G*Power 3.1 [56], with a 0.30 effect size, power of over 0.95, and an alpha level of 0.05, as being sufficient for both repeated measures, within–between interaction Analysis of Variance (ANOVA). The minimum number of participants required is 48 (16 participants for each group), but we have decided to reach a total number of 60 participants (20 participants for each group) in order to avoid possible drop-out effects.

For the purposes of this study, different types of statistical analysis will be carried out using SPSS 18 (SPSS Inc., Chicago, IL, USA) and a significant threshold throughout this study will be set at $p < 0.05$.

Statistical analysis will be of two types:

- "Within-group" analyses. These statistical analyses will be aimed at evaluating any improvement in patients' cognitive performance. Within each group of patients, the assessments at T0 will be compared with that at T1 and T2 through a repeated measures ANOVA between the scores of the cognitive assessments. In case of significance, post hoc tests will be carried out to which a Bonferroni correction will be applied with significance set at $p < 0.016$. Similarly, the same analysis will be conducted to compare the levels of anxiety and depression, the self-perception of cognitive deficit and the level of perceived disability.
- "Between-group" analyses. These statistical analyses will be aimed at comparing the results obtained by the groups at T0, T1, and T2 by performing an ANOVA between the scores of the cognitive assessments. In case of significance, post hoc tests will be performed to which a Bonferroni correction will be applied with significance set at $p < 0.016$. Additionally, in this case, the same comparative analysis will be carried out on the following variables: levels of anxiety and depression, self-perception of one's cognitive difficulties and level of perceived disability.

### 4.2. Outcomes

The primary outcomes of this study are the scores on the cognitive tasks and the scores on the gait and balance efficiency scales.

The secondary outcomes are the scores on the self-perception of cognitive deficits scale and on the self-perceived quality of life through the WHO-DAS.

The outcomes will be considered at T1 and T2.

## 5. Ethics and Dissemination

This study involving human participants has been reviewed and approved by I.R.C.C.S "Santa Lucia" Foundation Ethics Committee (CE/PROG. 698; 26 July 2018). All the participants will provide written informed consent to participate in this study.

## 6. Discussion

The present study will have the benefit of comparing three rehabilitation training sessions with a specific focus on cognitive outcomes. To date, the possible option for the management of cognitive dysfunctions in MS is CR, applied separately or as a part of a multimodal program involving also MR. However, in clinical practice, multimodal programs are very infrequent and patients with MS are primarily treated via pharmacological approach and subsequently via motor exercise training (physiotherapy). CR still remains poorly accessible for MS patients.

On the other hand, dual-task studies have shown how motor and cognitive skills are closely associated [57,58]. For example, during the execution of specific motor exercises, the subject automatically activates the processes of planning, control and coordination (executive functions) which, in turn, we know are connected to memory functions. It is, therefore, possible that physiotherapy exercises favor an improvement in memory performance indirectly, by strengthening the executive functions associated with memory.

In fact, the SNC should not be thought of as a set of independent modules, but as a network or a set of networks closely linked to each other. The brain areas are richly connected from an anatomical and functional point of view: therefore, by activating, for example, the motor cortex with physiotherapy exercises, it cannot be excluded that neuronal activity in the involved brain areas may indirectly enhance memory functioning. Therefore, we should consider motor and cognitive skills only apparently independent.

In this regard, one study reported significant memory improvements and concomitant increases in hippocampal volume following aerobic training in two people with MS and memory impairment [59]. On the other hand, the execution of specific memory exercises could reduce motor disabilities by the indirect action on the relative brain circuits.

As a consequence, we could expect that the combined rehabilitation (Reha 2) would allow a double advantage (both at a cognitive and motor level), thus ensuring greater effectiveness for the patient and a lower amount of time and resources.

Therefore, through this study, we aim to verify the hypothesis according to which combined rehabilitation intervention may be the best strategy to be used in MS patients having both motor and cognitive disabilities. We expect that a combined rehabilitation approach can bring a wider advantage as it may allow the enhancement of motor and cognitive skills at the same time.

Finally, by comparing the three rehabilitation approaches, we could even expect to highlight how the combined rehabilitation intervention favors an improvement in memory performance even higher than the CR alone.

If these results are reached, a change in clinical practice could be suggested: the combination of the two approaches in the same training with a reduction in time and costs for national health systems and a double advantage for MS patients.

**Supplementary Materials:** The following supporting information can be downloaded at: https://www.mdpi.com/article/10.3390/neurosci3030029/s1, Motor Rehabilitation Training.

**Author Contributions:** O.A., conceptualization, project administration, and writing—original draft preparation. C.P., investigation, data curation, and writing—review and editing. M.B., patient recruitment and assessment, and writing—review and editing. C.C., funding acquisition and writing—review and editing. A.S., writing—review and editing. V.S., funding acquisition and writing—review and editing. U.N., conceptualization and writing—review and editing. All authors have read and agreed to the published version of the manuscript.

**Funding:** This research was funded by the "Turati" Foundation with a one-year direct scholarship to O.A (Scholarship grant FTO.1) and by the Italian National Health System (SSN; SS 2018-2020 Clinical Neuroscience and Neurodegeneration—CNN; Brain Disorders and Clinical Neuroscience-IRG).

**Institutional Review Board Statement:** This study has been conceptualized according to the guidelines of the Declaration of Helsinki, and approved by the Ethics Committee of I.R.C.C.S. "Santa Lucia" Foundation (protocol code CE/PROG.698; 26 July 2018).

**Informed Consent Statement:** Informed consent has been prepared and will be signed from all subjects involved in this study.

**Data Availability Statement:** Not applicable.

**Conflicts of Interest:** V.S. and A.S. are part of the Scientific Direction of "Turati" Foundation who will contribute with one year scholarship of the first author of this study. The other authors declare no conflict of interest.

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
