# Peer review of "Study Protocol: Strategies and Techniques for the Rehabilitation of Cognitive and Motor Deficits in Patients with Multiple Sclerosis"

_neurosci, doi:10.3390/neurosci3030029_

Round 1
Reviewer 1 Report
Dear Authors, I read with interest your paper. I only have few minor objections:
-Figure 1 seems to be written in Italian language. I suggest translate it.
-some sentences throughout the paper are hard to read. I suggest an extensive English check language.
-finally I would further elaborate background particularly lines 59-66.
Once these changes are made the paper can be published to increase knowledge in the field.
Author Response
Reviewer 1
- My co-authors and would really thank the reviewer for his/her careful revision. We have tried to solve all the issues raised in order to improve the paper and make it suitable for publication.
-Figure 1 seems to be written in Italian language. I suggest translate it.
- We would like to thank the reviewer for his careful work. As suggested we translated the information in Figure 1.
-some sentences throughout the paper are hard to read. I suggest an extensive English check language
- We have carefully revised the manuscript for English language as suggested by the reviewer.
-finally I would further elaborate background particularly lines 59-66.
- We have modified the paragraph indicated by the reviewer adding more information, as suggested.
Once these changes are made the paper can be published to increase knowledge in the field.

Reviewer 2 Report
I reviewed the study protocol entitled" Study protocol: strategies and techniques for the rehabilitation 2 of memory deficits in patients with multiple sclerosis." It is a well-writen protocol for a valuable upcoming research study. However, there are some points that should be addressed:
1- The abstract demonstrates a few data about the methods of the project like the inclusion and exclusion criteria, questionnaires that will be used, and so on. It is better to be more comprehensive and include all parts of the protocol.
2- It is not defined that which statistical software will be used in this project.
3- The title is better to be more inclusive than just memory improvement, as the authors are going to do a cognitive rehabilitation, than a memory training.
4- Figure-1 is bettor to show more exact image of the whole study on the whole patients, including the tests that are taken and also the status of the control group.
Author Response
Reviewer 2
- My co-authors and would really thank the reviewer for his/her careful revision. We have tried to solve all the issues raised in order to improve the paper and make it suitable for publication.
1- The abstract demonstrates a few data about the methods of the project like the inclusion and exclusion criteria, questionnaires that will be used, and so on. It is better to be more comprehensive and include all parts of the protocol.
- The abstract has been changed and more details about the methods have been added, following reviewer suggestion.
2- It is not defined that which statistical software will be used in this project.
- As suggested by the reviewer we included this information in 4.1 Data analysis section
3- The title is better to be more inclusive than just memory improvement, as the authors are going to do a cognitive rehabilitation, than a memory training.
- The title has been changed according to reviewer suggestion and has been made more inclusive considering cognitive and motor deficits in general and not just memory deficits.
4- Figure-1 is better to show more exact image of the whole study on the whole patients, including the tests that are taken and also the status of the control group.
- The figure has been correctly renumbered and ameliorated according to reviewer suggestions.
